# Post Seismic Catalog Incompleteness and Aftershock Forecasting

**Eugenio Lippiello [1],\* , Alessandra Cirillo [1], Cataldo Godano [1] and Elefetheria Papadimitriou [2] and Vassilis Karakostas [2]**

[1] Department of Mathematics and Physics, University of Campania "L. Vanvitelli", Viale Lincoln 5, 81100 Caserta, Italy

[2] Geophysics Department, Aristotle University of Thessaloniki, GR 541 24 Thessaloniki, Greece

\* Correspondence: eugenio.lippiello@unicampania.it

**Abstract:** A growing interest appears among public authorities and society in accurate and nearly real time aftershock forecasting to manage and mitigate post-seismic risk. Existing methods for aftershock forecasting are strongly affected by the incompleteness of the instrumental datasets available soon after the main shock occurrence. The deficit of observed events, in the first part of aftershock sequences, can be naturally attributed to various mechanisms such as the inefficiency of the seismic network and the overlap of earthquake signals in seismic records. In this review, we show that short-term aftershock incompleteness can be explained only in terms of the second mechanism, whereas it is only weakly affected by the quality of the instrumental coverage. We then illustrate how standard models for earthquake forecasting can be modified to take into account this incompleteness. In particular, we focus on forecasting methods based on the data available in real time, in which many events are missing and the uncertainty in hypocenter location is considerable. We present retrospective tests that demonstrate the usefulness of these novel methods compared with traditional ones, which implement average values of parameters obtained from previous sequences.

**Keywords:** catalog incompleteness; seismic hazard

## 1. Introduction

Even if a still unanswered question is whether or not the accurate, reliable prediction of individual earthquakes is a realistic scientific goal, the possibility of forecasting future earthquakes exists. The two major examples concern the estimation of the occurrence probability of large shocks over a very long temporal interval (decades up to centuries) and the estimation of the aftershock occurrence rate after a large earthquake. Neither of the two cases is relevant in predicting the occurrence of an impending large earthquake but both examples provide very useful information on mitigating the impact of earthquakes that are likely to occur. As a matter of fact, the first example, usually defined as long-term (LT) seismic forecasting, is probably the most relevant from an engineering point of view, such as urban planning and building constructions: It allows one to address questions such as the maximum magnitude expected in a given area for the next years. Concerning the second example, usually defined as post-seismic Short-Term Aftershock (STA) forecasting, many events (the aftershocks) are always observed soon after the occurrence of a strong shock (the main shock). Aftershocks can attain sizes comparable to their triggering mainshock and can be very dangerous since they impact buildings already damaged by the previous shocks.

This review is focused on STA forecasting that can be potentially very efficient. Indeed the organization in time, space and energy of aftershocks follows well established empirical laws such as the Gutenberg–Richter (GR) and the Omori–Utsu (OU) law [1,2], which can be implemented in forecasting models. The GR law states that the magnitude distribution of earthquakes is an exponential function $P(m) \sim \exp(-\beta m)$, and the OU law characterizes the power law decay of the aftershock rate as function of the time $t$ since the main shock.

Even if the LT and STA forecasting act on two very different time scales, the two problems are intimately related. In the most simple description, seismic occurrence can be viewed as the superposition of two different stochastic processes: background seismicity responsible for mainshocks, which are the target of the LT forecasting, and aftershock occurrence, which is the target of STA. Hence, to achieve an accurate LT forecasting method a so-called declustering procedure is necessary, which allows one to isolate the two processes by means of a detailed knowledge of aftershock features. A clear example is the Epidemic Type Aftershock Sequence (ETAS) model introduced by Ogata [3] and probably representing nowadays the most popular model for STA as well as among the most efficient tools for LT forecasting. Studies of STA forecasting models, such as the ETAS model or more simple models implementing the OU law, have shown [4–14] that the incompleteness of datasets strongly affects the estimation of model parameters. This effect is more relevant in the first part of aftershock sequences when many earthquakes, in particular small ones, are not recorded and therefore not reported in seismic catalogs. This is mainly caused by the overlap of the signal of individual earthquakes in the seismic records. At the same time, incompleteness is also produced by the overload of processing facilities, due to a very large number of events in a narrow temporal window, and the damage caused by the mainshock to the seismic stations. Because of these difficulties, in many cases, operational probability forecasts only start more than 24 h after the mainshock [15].

In this review, we explore the problem of incompleteness of instrumental datasets focusing in particular on the so-called Short Term Aftershock Incompleteness (STAI). This is the main subject of Section 2. In Section 3, we review recent results on the influence of STAI on the estimation of parameters of STA forecasting models. Section 4 is then devoted to show that STAI is an intrinsic property of seismic catalogs which is not related to the efficiency of the seismic network. We conversely show that the main mechanism responsible for STAI is the overlap of aftershock coda waves with the waveforms of other events which obscure small aftershocks that occur close in time after larger ones. In Section 5, we show some approaches recently proposed to take explicitly into account this "obscuration" effect within the ETAS model. These approaches, however, are not simple to be implemented in real-time automatic procedures for aftershock forecasting. This is the topic of Section 6, which presents two different procedures developed to provide accurate STA forecasting, several minutes after the occurrence of a mainshock: the Omi et al. method [7,9,10] and the Lippiello et al. method [16,17]. The test of these two methods in retrospective studies is presented in Section 6 and final conclusions are drawn in the last section.

## 2. Catalog Incompleteness

Catalog completeness is usually quantified in terms of a magnitude threshold (or lower cut-off) $m_c$ defined as the magnitude above which all events are identified and included in the catalog. An accurate estimate of $m_c$ is fundamental in seismic forecasting. A too high value, discarding usable data, leads to loss information by under-sampling. Conversely, a too low value leads to an unreliable estimation of parameter values and thus to a biased analysis because of the incomplete dataset. A standard way of estimating $m_c$ is to find the minimum magnitude above which the best fit with the GR law is obtained. The value of $m_c$ clearly depends on the ability to filter noise and on the distance between the earthquake epicenter and the seismic stations necessary to trigger an event declaration in a catalog. Instrumental data from Taiwan seismicity, for example, give [18] at a given location $\vec{r}$, $m_c(\vec{r})$

$$m_c(\vec{r}) = 4.83d^{0.09} - 4.36. \tag{1}$$

where $d = |\vec{r} - \vec{r}_3|$ is the distance in kilometers between the epicenter and the position $\vec{r}_3$ of the third nearest seismic station. In Figure 1, we present the $m_c$ map for the Southern California obtained in [19] via the method of Amorése [20]. In particular, we observe a region in the central part of Southern California with a higher density of seismic stations, characterized by $m_c \leq 1.4$. This region, defined as Region 1, contains the 36% of $m > 2.5$ events recorded in the entire catalog. The remaining Southern California region (defined as Region 2) has a completeness magnitude starting from $m_c = 1.5$ and becoming as large as $m_c \simeq 3$ near the borders. A similar behavior is found if $m_c$ is evaluated according to the method of Schorlemmer and Woessner [21].

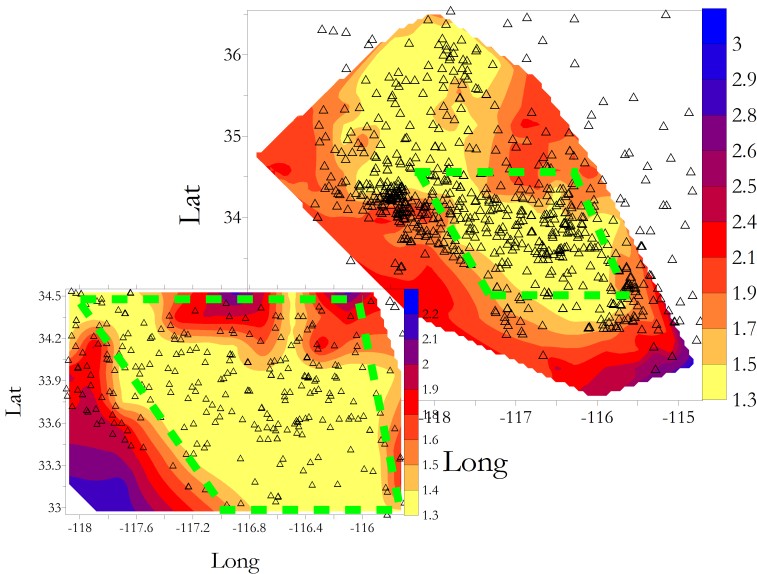

**Figure 1.** Magnitude completeness in Southern California. The value of $m_c$ can be obtained by the color bar and triangles identify the location of seismic stations. Green dashed lines define Region 1. Region 2 is the complement to Region 1 with respect to the entire Southern California (From [19]).

We stress that $m_c$ estimated from Equation (1) is a static quantity, controlled by the number of seismic stations, and we define it as "the static completeness magnitude". On the other hand, instrumental data show that the $m_c$ value, inside a given region, changes with time reaching much larger values in the first part of the aftershock sequence. As already anticipated in the Introduction, the dependence of the completeness magnitude $m_c(t)$ on the time $t$ since the main shock occurrence is usually termed Short Term Aftershock Incompleteness (STAI). Results in [22–24] give a completeness magnitude $m_c(t)$ which depends logarithmically on the time $t$ since the main shock

$$m_c(t, m_M) = m_M - \frac{1}{d}\left(\log_{10}\left(\frac{t}{C_0}\right)\right), \tag{2}$$

where $m_M$ is the main shock magnitude and $d$ and $C_0$ are fitting parameters. We refer to Equation (2) as the Kagan–Helmstetter formula with the best fitting parameters $d \simeq 1$ and $C_0 \sim 10^{-4}$ days, when time is measured in days. In Figure 2, we plot the experimental aftershock magnitude distribution evaluated for different temporal intervals after the $m = 7.3$ Landers earthquake, in Southern California. Experimental results show a magnitude distribution with an about flat for values $m < m_c(t)$, whereas curves appear parallel on a semi-logarithmic scale for $m > m_c(t)$ consistently with a GR law with $b \simeq 1$. The crossover magnitude $m_c(t)$ is in agreement with Equation (2).

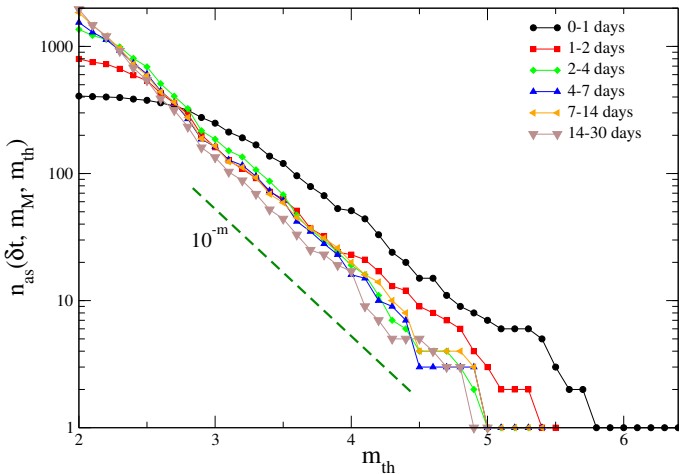

**Figure 2.** The number of aftershocks with magnitude larger than $m_{th}$ for the $m_M = 7.3$ Landers earthquake in Southern California, evaluated in different temporal windows $\delta t$ from the main shock. The green dashed line is the exponential behavior expected according to the GR law with $b = 1$.

In a different approach [5,7,9], STAI is taken into account by considering a magnitude distribution

$$P_{\beta,\sigma}(m) \propto e^{-\beta m}\Phi\left(m|\mu(t),\sigma\right) \tag{3}$$

given by the GR law multiplied by the detection rate function $\Phi$, which is represented by an error function

$$\Phi(m|\mu(t),\sigma) = \frac{1}{\sqrt{2\pi\sigma^2}} \int_{-\infty}^{m} e^{-\frac{(x-\mu(t))^2}{2\sigma^2}} dx. \tag{4}$$

In the above equation, the function $\mu(t)$ represents the 50% detection magnitude and $\sigma$ represents the range of the magnitude of partially detected earthquakes, i.e., at time $t$, only 50% of the events with $m = \mu(t)$ are expected to be detected whereas more the 98% of events are expected to be detected if $m > \mu(t) + 2\sigma$. A reasonable definition therefore corresponds to assume $m_c(t) = \mu(t) + 2\sigma$. In particular, Ogata and Katsura [5] proposed that $\mu(t)$ obeys the law

$$\mu(t) = \nu_0 + \nu_1 \exp\left(-\nu_2\left(3 + \log_{10}(t)\right)^{\nu_4}\right) \tag{5}$$

where the $\nu_i$ are fitting parameters. On the other hand, in a series of papers, Omi et al [7–10,15] developed an elegant method to obtain a non parametric fit of the function $\mu(t)$ and an estimate of $\sigma$ from the occurrence times and magnitudes of all recorded events in a giving learning period.

In Figure 3, we plot the results by Omi et al. [9] for $\mu(t)$ and $\mu(t) + 2\sigma$ for three aftershock sequences in Japan. These results are compared with the Ogata–Hirata formula (Equation (5)) and the Kagan–Helmstetter formula (Equation (2)). Figure 3 shows that the Omi and the Ogata–Hirata models give similar behavior for $\mu(t)$ and are able to capture the time variation of the detection rate. In contrast with these two models, since the parameters of the Kagan–Helmstetter formula are fixed for all sequences, it cannot reproduce the diverse recovering dynamics of the completeness magnitude that considerably depends on each aftershock sequence. The comparison of the forecasting skill of these three methods, for 38 Japan aftershock sequences, shows that the Omi method performs slightly better than the Ogata–Hirata methods and much better than a Kagan–Helmstetter formula [9].

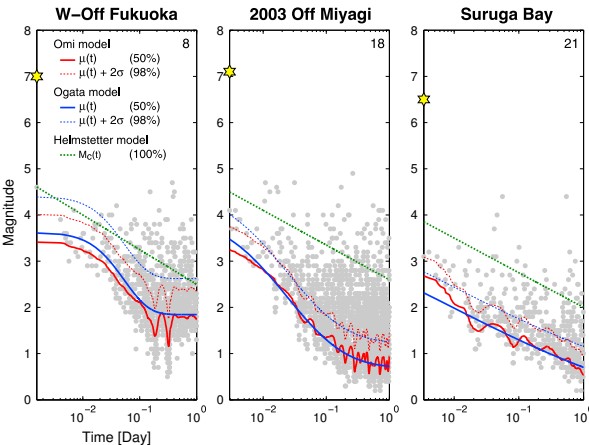

**Figure 3.** Examples of the estimated time-varying 50% detection rate $\mu(t)$ (solid curve) of magnitudes and estimated various time-varying completeness magnitudes (dotted curve) as indicated in the inset, superimposed on the magnitude-time plot of the observed aftershocks during the first day of the main shock. From [9].

## 3. The Influence of STAI on Model Parameters

For a complete dataset, one expects that the rate of aftershocks $\rho(t, m_M, m_{th})$ with magnitude larger than a threshold value $m_{th}$ occurring after a time $t$ following a mainshock of magnitude $m_M$ can be obtained by combining the GR law and the OU law

$$\rho(t, m_M, m_{th}) = \frac{K}{(t+c)^p} e^{-\beta m_{th}}. \tag{6}$$

According to the productivity law [25], $K$ depends on the main shock magnitude and Equation (6) can be written as

$$\rho(t, m_M, m_{th}) = \frac{K_0 e^{\alpha m_M - \beta m_{th}}}{(t+c)^p}. \tag{7}$$

As already observed in [2], missing small events in the early stage of the aftershock sequence causes the instability of the estimate of the parameters $K_0, \alpha, \beta, c, p$ in Equation (6). A problem which becomes particularly relevant at the beginning of aftershock sequences when the completeness magnitude after large earthquakes can temporarily increase by several units [4,22,26,27]. For this reason, long and short term forecasts usually present some corrections which take into account STAI [6,28,29].

Incompleteness, in particular, can make the $c$-value measured from instrumental catalogs $c_{meas}$ much larger the "true" $c$-value in the OU law (Equation (6)). Indeed, restricting to aftershocks with magnitudes larger than a reference value $m_{th}$, if events with magnitudes $m < m_c(t)$ are not recorded, the measured $c$-value can be obtained from Equation (2) after setting $m_c(c_{meas}) = m_{th}$, which leads to

$$c_{meas} = C_0 10^{d(m_M - m_{th})}. \tag{8}$$

It is evident that this quantity depends on the parameters of Equation (2) but is not related to the $c$-value of the OU law. Alternatively, an estimate of $c_{meas}$ can be obtained from Equation (5) after setting $\mu(c_{meas}) = m_{th} - 2\sigma$. As a consequence, the incompleteness at short times hides the true value of $c$ that in turn introduces a strong bias in the evaluation of the parameters $K_0$ and $\alpha$ in Equation (6), strongly affecting routines for short term aftershock forecasting at time $t < c_{meas}$.

### 3.1. The Influence of STAI on the ETAS Parameters

As anticipated in the Introduction, the ETAS model is probably, nowadays, the most popular one for STA forecasting. The assumptions of the ETAS model include: (1) yhe background seismicity

is a stationary Poisson process that depends on the position $\vec{x}$, $\mu(\vec{x})$; (2) every event, whether it is a background or a triggered one by a previous event, triggers its own off-spring independently; (3) the expected number of direct off-springs is an exponential function of the magnitude of the mother event (productivity law); and (4) the time lags between triggered events and the mother event follow the OU law. According to these assumptions, the occurrence rate of events with magnitudes $m \geq m_0$ at the position $\vec{x}$ at time $t$ is given by

$$\Lambda_{ETAS}(m, \vec{x}, t) \quad = \quad \left[ \sum_{i=1}^{N} Q\left(|\vec{x}_i - \vec{x}|, t - t_i, m_i\right) + \mu(\vec{x}) \right] \beta e^{-\beta(m - m_0)} \tag{9}$$

where the sum extends over all events with magnitude $m_i$, epicentral coordinate $\vec{x}_i$ and occurrence time $t_i < t$ and

$$Q(\Delta r_i, t - t_i, m_i) \quad = \quad \frac{K_0(p-1)}{c} e^{\alpha(m_i - m_0)} \left(1 + \frac{t - t_i}{c}\right)^{-p} G(\Delta r_i, m_i) \tag{10}$$

with $\Delta r_i = |\vec{x}_i - \vec{x}|$, which is the epicentral distance. The function $G(\Delta r_i, m_i)$ is a spatial kernel that explicitly depends on the triggering magnitude $m_i$ and $\mu(\vec{x})$ is the time independent contribution due to background seismicity.

The influence of STAI on the estimates of the ETAS parameter was addressed by Zhuang et al. [13] in the case of the 15 April 2016, Kumamoto earthquake sequence in Japan. Under the assumption that earthquake magnitudes are independent of their occurrence times, Zhuang et al. [13] replenished the short-term missing data of small earthquakes by using a bi-scale transformation. They then compared the maximum likelihood estimate of the ETAS parameters of the recorded dataset in the JMA catalog with the replenished one, considering only events above a lower magnitude threshold $m_{th} = m_c$. Results plotted in Figure 4, as function of $m_c$, show that, when the magnitude threshold $m_c \geq 3$, which is approximately the static completeness magnitude of the JMA catalog, the estimated ETAS parameters are about the same for both datasets. Conversely, important differences are found for values of $m_c < 3$. For the replenished dataset, the estimated background rate $\mu(x)$ decreases roughly exponentially when the cut-off magnitude is increased, consistently to what is expected according to the GR law ( Figure 4a). The  original dataset, conversely, exhibits a flatter behavior, indicating the absence of small magnitude events. Concerning the other parameters, the most striking feature is that in the replenished dataset all parameters only weakly depend on $m_c$, as expected, whereas we observe a non-trivial dependence on $m_c$ in the JMA catalog.

The results of Zhuang et al. [13] indicate that the estimate of ETAS parameters from the original dataset, when one considers a lower magnitude threshold $m_c < 3$, leads to non-correct results. A similar conclusion was reached by Seif et al. [14] who studied how the ETAS parameters, obtained by the iterative approach of Zhuang et al. [30], depends on the lower magnitude threshold $m_{th}$. In particular, Seif et al. [14] investigated two simulated ETAS catalogs: a complete one which implements the ETAS parameters estimated from the Southern California catalog and an incomplete one where aftershocks of mainshocks with $m_M > 5$ were removed if their magnitude was smaller than $m_c(t)$ given in Equation (2). Results plotted in Figure 5 show that for sufficiently larger values of $m_{th}$, the parameter inversion procedure does not give the true values of $K_0$ and $p$ used to generate synthetic catalogs. Seif et al. [14] attributed the observed discrepancy to the fact that aftershocks triggered by events with $m < m_{th}$ are erroneously identified as direct aftershocks of some previous larger earthquake. This widens the distribution of direct aftershocks leading to a smaller $p$-value. At the same time, because of the anticorrelation between $K_0$ and $p$, $K_0$ is overestimated. Figure 5, in particular, shows a striking difference between the estimated parameters in the complete and the incomplete catalogs. However, this difference tends to disappear for increasing $m_{th}$ indicating that the influence of aftershock incompleteness is not significant for $m_{th} \gtrsim 3.5$.

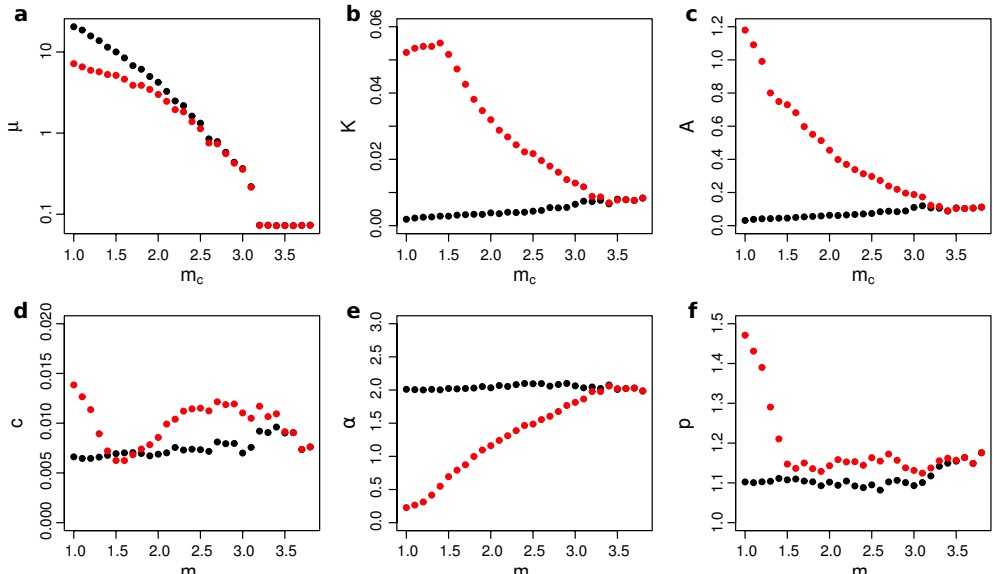

**Figure 4.** Different panels correspond to the ETAS parameters $\mu, K, A = K \int_0^\infty (t+c)^{-p} dt, c, \alpha, p$ (see axis labels) estimated from the Kumamoto aftershock sequence with different magnitude thresholds. The red and black dots are the estimates based on the original and the replenished datasets, respectively. Unit of measures are $day^{-1}, day^{p-1}, day$ for $\mu, K, c$ respectively and the other quantities are adimensional except $A = K \int_0^\infty (t+c)^{-p} dt$ which represents the productivity from an event of magnitude $m_c$. From [13].

Results of Figures 4 and 5 indicate that using a lower magnitude threshold $m_{th}$ below the completeness level, especially for some parameters, can lead to incorrect prediction. Unfortunately, it is not simple to establish a strict correspondence between the degree of incompleteness of the catalog and the error expected in the estimate of parameters.

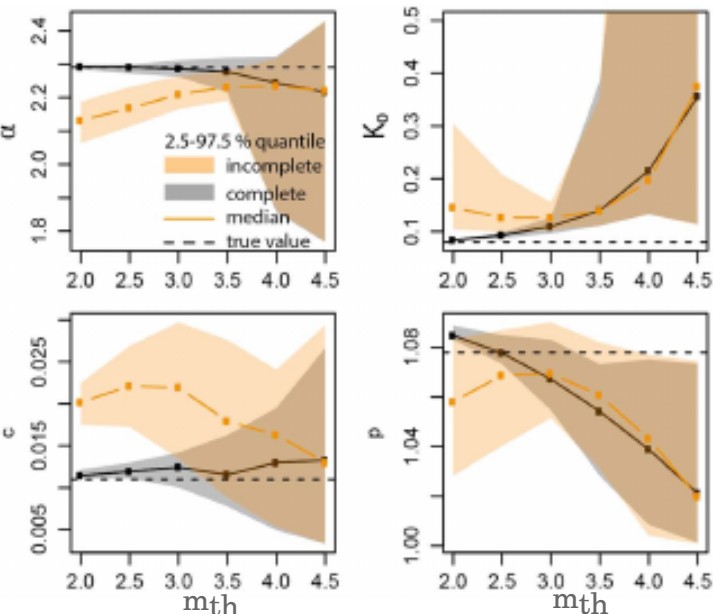

**Figure 5.** The ETAS parameters are plotted against $m_{th}$ for synthetic catalogs simulated with parameters from Southern California (gray) and compared with the parameter for the incomplete synthetic catalog (orange). The "true" parameter values are plotted with black dashed lines the grey shadowed region represents the 95% quantiles of 30 synthetic ETAS catalogs. The orange shadowed region represents the 95% quantiles of 30 synthetic ETAS incomplete catalogs. From [13].

### 3.2. Is STAI Related to the Static $m_c$?

As explained in Section 2 the static $m_c$ is a local quantity which depends on the local density of the seismic network $\rho_S$, as illustrated by Equation (1) and Figure 1. The influence of the density $\rho_S$ on the STAI was addressed by de Arcangelis et al. [31] by investigating the $c_{meas}$-value in the two sub-regions of Southern California illustrated in Figure 1. As already explained in Section 2, the inner region (Region 1) comprises a high value of $\rho_S$ and a static $m_c \leq 1.4$. Conversely, a small $\rho_S$ is present in the external region (Region 2) and the static $m_c > 1.5$, with values of $m_c \simeq 3$ close to the borders. To obtain an estimate of $c_{meas}$ in each sub-region, de Arcangelis et al. [31] measured the aftershock daily rate $\rho(t, m_M, m_{th})$ defined as the number of aftershocks with magnitude larger than $m_{th}$ occurring at a temporal distance $t$ after their triggering main shock with magnitude $m \in [m_M, m_M + 1)$, divided by the number of mainshocks with magnitude $m \in [m_M, m_M + 1)$. Three different values of $m_M = (3, 4, 5)$ and $m_{th} = (1.5, 2.5, 3.5)$ were considered. In this study, mainshock–aftershock couples were identified according to the Baiesi–Paczusky (BP) declustering criterion [32–34] using the same parameters adopted by Moradpour et al. [35] and Hainzl [12]. In particular, only aftershocks identified as direct descendants of the mainshock were included in the analysis.

The results (Figure 6) show that the aftershock rate clearly depends on the magnitude difference $m_M - m_{th}$ in both Region 1 and Region 2. In particular, de Arcangelis et al. [31] divided time by $\tau = 10^{d(m_M - m_{th})}$ obtaining that data for different values of $m_M$ and $m_{th}$, inside each sub-region, exhibits the scaling collapse $\rho(t, m_M, m_{th}) = F(t/\tau)$ (Figure 7a). It is evident from Figure 7a that the Omori decay $\rho \sim t^{-p}$ sets in when $t/\tau$ becomes larger than a given value $x_0$, different between the two regions. Since the $c_{meas}$ can be obtained from the time such that the Omori decay $\rho \sim t^{-p}$ sets in, Figure 7a gives $c_{meas} = x_0\tau$ and one recovers Equation (8) after the identification $x_0 = C_0$. In particular the best fit gives $\log_{10}(C_0) = -3.53 \pm 0.05$ and $d = 1 \pm 0.03$ inside Region 1 and $\log_{10}(C_0) = -3.70 \pm 0.05$ and $d = 0.95 \pm 0.03$ inside Region 2. This leads to a counterintuitive behavior with a $c_{meas}$-value being larger inside Region 1 even if the static $m_c$ is significantly smaller inside Region 1 than in Region 2. Conversely a smaller $c_{meas}$-value is found in Region 2 when the static $m_c$ is larger. This result clearly indicates that $c_{meas}$ is not related to $\rho_S$ and that STAI cannot be reduced by increasing the density of the seismic station thus suggesting that STAI originates from a different mechanism (see next section). The same conclusion can be also obtained from the measurement of the correlation between magnitude according to the method proposed in [19,36–38]. This analysis [19,31] has shown significantly larger magnitude correlations in Region 1 than in Region 2.

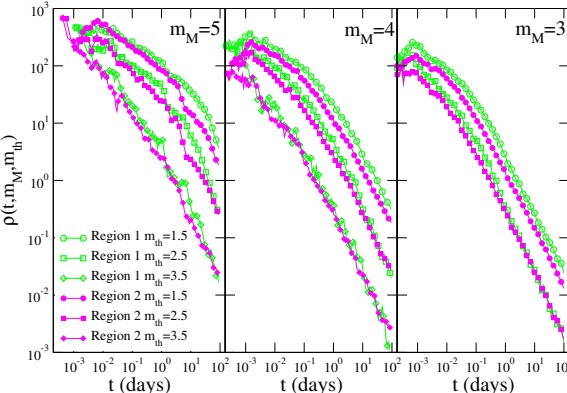

**Figure 6.** The number of events identified as aftershocks by the BP declustering procedure with magnitude larger than $m_{th}$, which occurred at a temporal distance $t$ from events identified as mainshocks with magnitude $m \in [m_M, m_M + 1)$, is divided by the number of identified mainshocks and plotted versus $t$. Different panels correspond to different values of the mainshock magnitude class $m \in [m_M, m_M + 1)$. Different colors correspond to results for different geographic regions: Region 1 (open green symbols) and Region 2 (filled magenta symbols). Different symbols indicate different values of the lower threshold: $m_{th} = 1.5$ (circles), $m_{th} = 2.5$ (squares) and $m_{th} = 3.5$ (diamonds).

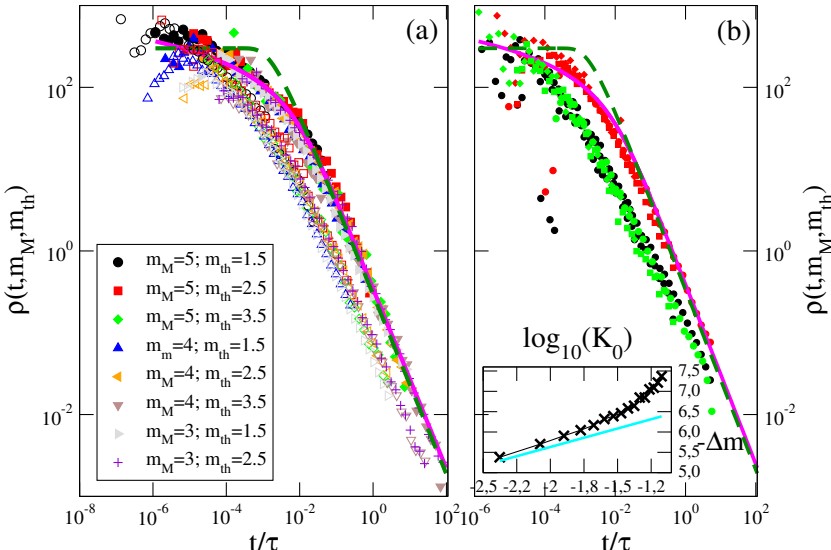

**Figure 7.** (Color online) (**a**) The same data in Figure 6 are plotted as function of $t/\tau$, with $\tau = 10^{d(m_M - m_{th})}$ proportional to $c_{meas}$ (Equation (8)) with $d = 1$, for different values of $m_M$ and $m_{th}$. Filled (empty) colored symbols are used for data of Region 1 (Region 2). The magenta continuous line is the scaling function $F(x) = A\log\left(1 + Bx^{-p}\right)$ with $A = 0.35$, $B = 70$ and $p = 1.1$, whereas the dashed green line is the scaling function $F(x) = A(x/B + 1)^{-p}$ with $A = 300$, $B = 7$ and $p = 1.1$. (**b**) The aftershock density $\rho(t, m_M, m_{th})$ in the ETASI1 catalog, with a blind time $\Delta t = 1$ min, is plotted as a function of $t/\tau$. Different values of $m_M$ and $m_{th}$ are plotted with different symbols: stars for $m_M - m_{th} = 2.5$, crosses for $m_M - m_{th} = 1.5$ and plus for $m_M - m_{th} = 0.5$. Different colors correspond to different values of $K_0$ and of the average background rate $r_B$: $K_0 = 0.035$ and $r_B = 4.38$ days$^{-1}$ (black), $K_0 = 0.035$ and $r_B = 8.3$ days$^{-1}$ (green) and $K_0 = 0.068$ and $r_B = 4.38$ days$^{-1}$ (red). Magenta continuous and green dashed lines are the same scaling functions $F(x)$ plotted in (**a**). (Inset) The value of $\Delta m$ (Equation (8)) as function of $log_{10}(K_o)$ for the ETASI model with a blind time $\Delta t = 1$ min (black crosses). The cyan line is the theoretical prediction (Equation (18)). From [31].

## 4. The Origin of STAI and the Envelope Function

Results of the previous Section (Section 3.2) suggest that STAI is an intrinsic property of seismic catalogues not related to density of the seismic stations. This conclusion is strongly supported by the study of the envelope function $\mu_e(t)$ after several mainshocks that occurred in Greece and Italy in the last ten years [16]. More precisely, the envelope function $\mu_e(t)$ is obtained from the ground velocity recorded during the first days after the mainshock. The signal of each component is filtered by means of a two-pass Butterworth filter in the range $[1, 10]$ Hz, the envelope of each signal is computed and the signals of the three components are superimposed. $\mu_e(t)$ is finally defined as the logarithm of the resulting signal. This quantity was introduced by Peng et al. [26] to identify aftershocks not reported in the JMA catalog during the first minutes after the main shock. The idea is that the occurrence of an aftershock must produce a double peak in $\mu_e(t)$ corresponding to the coupled pair of P and S arrivals. The local magnitude of the event is given by $m \simeq \mu_{max} + const$, where $\mu_{max}$ is the maximum in $\mu_e$ and the constant depends on the epicentral distance from the recording station, related to the S-P time difference.

Considering the evolution of $\mu_e(t)$ after a mainshock, occurred at the time $t_0$, Lippiello et al. [16] found that the envelope function never goes below a given value $\mu_{min}(t)$ which is a logarithmic decreasing function of time (Figure 8)

$$\mu_{min}(t) = \mu_M - \phi \log(t - t_0) - \Delta\mu_{min}. \tag{11}$$

As a consequence, even very accurate analyses of post seismic waveforms, even those which employ sophisticated matched filter detection algorithms [39,40], do not allow one to identify small events which produce peaks smaller than $\mu_{min}(t)$. This reflects a completeness magnitude $m_c(t)$ that depends on the time after the mainshock with a functional dependence similar to $\mu_{min}(t)$ and, therefore, small events cannot found and catalogs are intrinsically incomplete.

To understand the mechanism responsible for the existence of $\mu_{min}(t)$, a closer inspection of the envelope function $\mu(t)$ after all mainshocks reveals the existence of two characteristic times: $\tau$ and $t_M$. The first time $\tau$ is of the order of some seconds, whereas $t_M$ is of order of some minutes, and three distinct regimes are observed:

- For $t - t_0 < \tau$, $\mu_e(t)$ increases to a maximum value $\mu_M$.
- For $\tau < t - t_0 < t_M$, $\mu_e(t)$ follows a logarithmic decay as

$$\mu_e(t) \simeq \mu_M - q \log(t - t_0). \tag{12}$$

- For $t - t_0 > t_M$, the average value of the envelope $\langle \mu_e(t) \rangle$ is still logarithmic but with different coefficients:

$$\langle \mu_e(t) \rangle = \mu_M - \phi \log(t - t_0) - \Delta\mu, \tag{13}$$

with $\phi < q$.

The same three regimes have been found for other mainshocks in Southern California and in Italy [16]. The first two regimes can be easily associated to the mainshock waveform, which can be modeled as $\mu_e(t - t_0) = \mu_M + \log[g(t - t_0)]$, where $g(t - t_0)$ is the mainshock envelope waveform. Experimental results suggest an initial linear increase of $g(t)$ [41] followed by a fast decay consistent with an exponential function $g(t) \sim \exp(-Q^{-1}t)$ [42]. Figure 8 indicates that in the intermediate regime $\tau < t - t_0 < t_M$, with $t_M$ of the order of few minutes, the envelope waveform is more consistent with a power law decay as proposed by Lee et al. [43]. Under these assumptions, the behavior of $g(t)$ up to the time $t - t_0 < t_M$ can be modeled as $g(t) \sim t(t/\tau + 1)^{-1-q}$ with the time $\tau$ representing the typical duration of the mainshock signal, leading to

$$\mu_e(t) = \mu_M + \log(t - t_0) - (q + 1)\log\left((t - t_0)/\tau + 1\right). \tag{14}$$

The existence of the third regime, previously enlightened by Sawazaki and Enescu [44], can be interpreted taking into account that not only the main shock but each aftershock of magnitude $m_i$, occurred at time $t_i$, produces a signal following the relation $\mu_e(t) = \mu_i + \log[g(t - t_i)]$ and one therefore expects a theoretical envelope of the form

$$\mu_{th}(t) = \log\left\{\max_{t_i < t}\left[10^{\mu_i} g(t - t_i)\right]\right\}, \tag{15}$$

where the maximum must be evaluated for all aftershocks with occurrence times $t_i < t$.

*Numerical Generation of the Envelope Function*

To verify that Equation (15) reproduces the experimental findings, Lippiello et al. [16] started from a mainshock with magnitude $m_M$ occurring at time $t_0$ and assumed that the aftershock rate follows the OU law (Equation (6)). Since $p$-values usually have small fluctuations among different aftershock sequences [45], Lippiello et al. [16] assumed a fixed value of $p$ ($p = 1.1$) and after choosing different values of $K$ and $c$, they generated an aftershock sequence according to Equation (6) for a temporal window of three days. To each aftershock is then associated a magnitude randomly extracted from the GR law. After fitting the value of $\tau$ from the experimental $\mu_e(t)$, the key assumption is that a magnitude $m_i$ aftershock, occurring at time $t_i$, generates a seismic signal with envelope $A(t) = 10^{m_i} g(t - t_i)$ with $g(t) = t(t/\tau + 1)^{-1-q}$ and $q = 2.5$. The synthetic $\mu_{th}(t)$ is then obtained from Equation (15) and a

vertical shift is finally applied in order to have the mainshock peak in $\mu_{th}(t)$ equal to the experimental $\mu_M$. The numerical parameters $K, c$, implemented in the OU law (Equation (6)) are then tuned in order to reach a good agreement between $\mu_{th}(t)$ and the experimental $\mu_e(t)$, according to the procedure described in Section 6.2. Results of $\mu_{th}(t)$ plotted as orange lines in Figure 8 show that it is possible to generate a synthetic envelope reproducing the experimental one in all the three regimes. The above results indicate that since each aftershock produces its own coda waves which decay as a power law with exponent $q$, the overlap of coda waves generated by subsequent aftershocks causes the existence of a lower signal $\mu_{min}(t)$ which decays as a power law with an exponent $\phi < q$ (Equation (13)). The same agreement between $\mu_e(t)$ and $\mu_{th}(t)$ is recovered for other mainshocks $m_M > 6$ recorded in Greece, Italy and Southern California [16].

We wish to stress that the mainshock peak $\mu_M$, as well as aftershock peaks $\mu_i$ in Equation (15), strongly depends on the distance of the recording station from the mainshock epicenter and on site effects. In addition, the functional form of $g(t)$ can be different at different stations. As a consequence both $\mu_e(t)$ and $\mu_{th}(t)$ are different at different stations but, under the hypothesis that aftershocks occur not too far from the mainshock hypocenter, the values of $K$ and $c$ providing the best agreement between $\mu_e(t)$ and $\mu_{th}(t)$ should be the same for all stations.

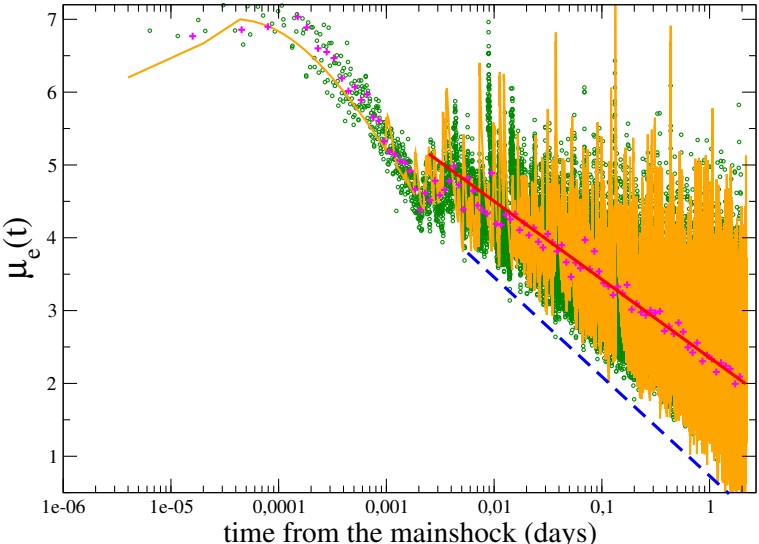

**Figure 8.** The quantity $\mu_e(t)$ (green circles) after the the Hector Mine earthquake in California recorded at the station CIGSC located at a distance of 92 km from the main shock epicenter. The magenta crosses indicate the (logarithmically binned with bin value 0.1) average value of $\mu(t)$, the red continuous lines represent the results of the logarithmic fit (Equation (13)) for $t - t_0 > t_M$. The dashed blue lines represent the quantity $\mu_{min}(t)$ and orange lines are used for results of numerical simulations for the theoretical envelope $\mu_{th}(t)$, defined in Equation (15). The values of the best-fitting parameters in Equation (6) are $K = 0.95, c = 0.18$ days and $\tau = 8$ s.

## 5. The ETASI Model

In the previous section, we have shown that STAI is mostly due to the overlap among aftershock coda waves. This ingredient can be incorporated in the ETAS model by multiplying the ETAS occurrence rate $\Lambda_{ETAS}$ in Equation (9) by a detection function

$$\Lambda_{ETASI}\left(\vec{x}, t, m | \vec{x}_i, t_i, m_i\right) = \Lambda_{ETAS}\left(\vec{x}, t, m | \vec{x}_i, t_i, m_i\right) \times \Phi(m, t, \mu(t) | m_i, t_i). \tag{16}$$

The detection rate can be still described by an error function as in Equation (4) and we define the model described by Equation (16) as the ETAS Incomplete (ETASI) model. The main difference with Equation (3) is that in this approach the detection function $\Phi(m, t, \mu(t) | m_i, t_i)$ depends on the history

of all previous earthquakes $\{m_i, t_i\}_{i=1}^{N}$, with $t_i < t$. More precisely, in Equation (3), the 50% detection function $\mu(t)$ depends only on the time and magnitude of the main shock whereas in Equation (16) each event can obscure the recording of subsequent earthquakes.

We observe that the ETASI model differs form the procedure adopted by Seif et al. [14] who generated incomplete ETAS catalogs by removing only aftershocks of mainshock with $m > 5$. In the ETASI model, conversely, any event can obscure subsequent earthquakes independently of its magnitude.

The simplest choice for the detection function $\Phi(m, t | m_i, t_i)$ was proposed by Hainzl [12] and corresponds to an error function with $\sigma \to 0$ and $\mu(t) = m_i$ if $t - t_i \leq \Delta t$, whereas $\mu(t) = 0$ for $t - t_i > \Delta t$, where $\Delta t$ is a constant blind time. This corresponds to the hypothesis that each earthquake hides all subsequent smaller events occurring at temporal distances smaller than $\Delta t$. Notwithstanding the simplicity of this functional form of $\mu(t)$, as already proposed by Hainzl [12], this model, defined as ETASI1 in the following, leads to non-trivial temporal patterns of the aftershock occurrence.

The hypothesis of a constant blind time allows one to achieve an analytical evaluation of $c_{meas}$ [12]. Indeed, the blind time $\Delta t$ also represents the minimum temporal distance between two subsequent earthquakes reported in a catalog and this leads to a maximum detectable rate $\rho_{max} \simeq 1/\Delta t$. As a consequence, since the "true" aftershock rate is a decreasing function of the time $t$ after the mainshock occurrence (Equation (6)), the measured $\rho(t, m_M, m_{th})$ corresponds to the "true" aftershock rate only if $\rho(t, m_M, m_{th}) < \rho_{max}$, a condition which is always fulfilled at large times. Conversely, at small times, when the "true" aftershock rate is larger than $\rho_{max}$, the measured $\rho$ exhibits a constant behavior $\rho(t, m_M, m_{th}) \simeq \rho_{max}$. Accordingly, the $c_{meas}$-value can be identified as the time such as $\rho(c_{meas}, m_M, m_{th}) = \rho_{max}$, and assuming $\alpha \simeq b$ Equation (7) gives

$$\rho(c_{meas}, m_M, m_{th}) = \frac{K_0 e^{b(m_M - m_{th})}}{(c_{meas} + c)^p} = \rho_{max}, \tag{17}$$

giving $c_{meas} = c + (K_0/\rho_{max})^{1/p} \exp(b/p)(m_M - m_{th})$, which for $c \ll c_{meas}$ coincides with Equation (8):

$$C_0 = \Delta m = \left(\frac{K_0}{\rho_{max}}\right)^{1/p}, \tag{18}$$

and $d = b/p$.

The ETASI1 model can be implemented numerically via a two step process. At the first step, standard ETAS catalogs are simulated and, at the second step, all events that occurred at a temporal distance smaller than $\Delta t$ after a larger event are removed from the catalog. de Arcangelis et al. [31] implemented different values of $K_0$ and analyzed the ETASI1 catalog by the same BP declustering procedure applied to the instrumental catalog. As in Figure 6, the aftershock daily rate $\rho(t, m_M, m_{th})$ for the ETASI1 catalog has been evaluated for different mainshock magnitudes $m_M$, different thresholds $m_{th}$ and different $K_0$ values. This study has shown that the $c_{meas}$-value follows Equation (8) with $d = b/p$ as illustrated in Figure 7b where $\rho(t, m_M, m_{th})$ is plotted as a function of $t/\tau$ with $\tau = 10^{d(m_M - m_{th})}$ and $d = b/p$. Data for different $m_M$ and $m_{th}$ and the same $K_0$ collapse onto the same master curve $F(t/\tau)$, as for the instrumental catalog (Figure 7a). Concerning the value of $C_0$, de Arcangelis et al. [31] observed that the larger the value of $K_0$ implemented in ETAS simulations the larger was the value of $C_0$ fitted from the decay of $\rho(t, m_M, m_{th})$. Results plotted in the inset of Figure 7b show that $-\Delta m = \log_{10} C_0$, becomes more positive for increasing $K_0$ confirming the strong correlation between $C_0$ and $K_0$. In particular, we observe that the dependence of $C_0$ on $K_0$ is consistent with Equation (18) only for small values of $K_0$. Deviations from Equation (18) can be attributed to the cascading process implemented in the ETAS model. Indeed, aftershocks of higher order generation are also followed by a blind time which eventually hides aftershocks of previous generations. This causes a larger total blind time compared to the situation when higher order generation aftershocks are not considered, as in Equation (18).

The comparison between the data collapse observed for the ETASI1 catalog (Figure 7b) with the one observed for the instrumental Southern California catalog (Figure 7a) suggests that the larger value $c_{meas}$ inside Region 1 must be attributed to a larger productivity (larger $K_0$) of that region. This is in agreement with the behavior of $\rho$ (Figure 6) for times $t > c_{meas}$ when the "true" OU decay $\rho \sim K/t^p$ is expected. Indeed, it is evident that, when $t > c_{meas}$, $\rho$ in Region 1 is systematically larger than in Region 2.

We further observe that the scaling function $F(x)$ presents clear deviations from the OU prediction $F(x) \propto (x+1)^{-p}$ in the intermediate temporal regime. We attribute these deviations to the cascading process which can produce a more gradual decrease of the aftershock number from the initial plateau compared to the situation when higher order generation aftershocks are not taken into account [12]. A better fit for $F(x)$ in numerical and instrumental catalogs is provided by $F(x) = A \log \left(1 + Bx^{-p}\right)$ obtained by Lippiello et al. [46] under a dynamical scaling assumption [38,45,47–51].

### 5.1. ETASI2

A more refined expression for $\mu(t)$ within the ETASI model (Equation (16)) is proposed in [31] and corresponds to the so called ETASI2 model. The idea is that the 50% detection function follows the same decay of the envelope function of a single earthquake and according to Equation (12) this corresponds to the assumption that

$$\mu(t) = \max_{i:t_i<t} \left(m_i - q \log(t - t_i) - \delta_0\right),\tag{19}$$

where the maximum is evaluated over all events with magnitude $m_i$ occurred at time $t_i < t$. The model is numerically implemented in [31] taking for the detection rate function $\Phi$ an error function as in Equation (4) with $\sigma \to 0$. This corresponds to the two-step procedure illustrated in the previous section with the removal from the original ETAS catalog of all events with magnitude $m$ and occurrence time $t$ such that $m < \mu(t)$. A finite value of $\sigma$ is considered in [52].

In de Arcangelis et al. [31], the coefficient $q$ in Equation (19) is taken as a model parameter and its value has been tuned in order to achieve the best agreement between the organization of aftershocks in ETASI2 and instrumental catalogs. This study showed that the ETASI2 model provides a more accurate description of aftershock occurrence, with respect to the ETASI1 model, and in particular it better captures the correlation between subsequent magnitudes observed in instrumental catalogs. In particular the agreement between instrumental and ETASI2 catalogs is obtained by setting a $K_0$ value, in the ETASI2 simulations, significantly larger inside Region 1 of Southern California (Figure 1) than Region 2. As a consequence, de Arcangelis et al. [31] proposed that the value of $K_0$ which provides the best overlap between ETASI2 and instrumental catalogs can be interpreted as the best estimate for the true productivity coefficient $K_0$ in each region.

### 5.2. Dynamical Scaling ETAS Model

A model alternative to the ETASI has been proposed on the basis of a dynamical scaling relation between time and energy [19,36,38,46,47]. Within this hypothesis, different from the general assumption of the ETAS model [3,53,54], time and magnitude are not independent quantities but the magnitude difference fixes a characteristic time scale for aftershock rate relaxation. Deviations from the GR law are a natural consequence of this assumption with a completeness magnitude depending on time in agreement with what is observed in experimental data (Equation (2)). The study of the maximum likelihood [51] has shown that this method provides a more accurate description of the aftershock rate decay than the ETAS model.

## 6. Automatic Procedures for Short-Term Aftershock Forecasting

In this section, we present two methods which have been developed in order to provide real-time aftershock forecasting: The Omi method [7,9,10] and the Lippiello method [16,17]. The idea of

both methods is to extrapolate the parameters of the OU law, or more generally of the ETAS model, by means of an automatic procedure which uses the information available up to a time $T_2$ after the mainshock. The ETASI model, presented in the previous section, is not suitable for this purpose because it is not possible to apply a maximum likelihood estimation procedure to invert parameters. For the likelihood evaluation, indeed, one should have access to "obscured" events, an information by definition unavailable. In this section, we review the retrospective tests performed with the Omi and the Lippiello methods. Both tests consider the forecasting according to the OU law (Equation (6)) implementing the parameters $K$ and $c$ estimated for each individual mainshock sequences, according to the information available in real time. This forecasting is compared to a generic model where the parameters $K$ and $c$ are taken as average values over many sequences. The results show that the novel methods outperform the generic model.

*6.1. The Omi Method*

The Omi method, briefly illustrated in Section 2, has been implemented in a real-time system for automatic aftershock forecasting in Japan. A systematic test of the efficiency of the Omi method, using real-time seismic data, was performed by Omi et al. [10] on aftershock sequences of seven inland mainshocks with magnitudes $m \geq 7$ that occurred after the establishment of the Hi-net observation system. The Omi method is based on the evaluation of the parameters $K, p, c$ in Equation (6) using the information from an incomplete dataset, including only the recorded aftershocks. More precisely, Omi et al. [10] considered data in the learning period from two instrumental catalogs: the Hi-net and JMA catalogs. The results of this method are compared to a standard forecasting approach which uses fixed parameter values (the generic model) determined based on many aftershock sequences in Japan. The forecast from the generic model depends only on the main shock magnitude. The performance is compared by means of the log-likelihood ratio score, which is referred to as information gain $I$. The standard error $S_I$ of the information gain is also numerically evaluated and, under a Gaussian approximation, one forecast performs better than the other one, with a probability larger than the 95%, if $I > 1.64S_I$. More precisely, Omi et al. [10] considered four learning periods corresponding to the first 3, 6, 12, and 24 h periods of aftershock data to prepare forecasts for the following 3, 6, 12, and 24 h testing periods, respectively. The results of the test, for the seven Japan aftershock sequences, are visually represented in Figure 9 that shows the information gain per aftershock, considering separately data from the Hi-net and JMA catalog, against the generic aftershock model. The error bars correspond to $1.64S_I$ and, therefore, if their lower bound is greater than zero, the Omi model performs better than the generic model. Omi et al. [10] separately considered two target magnitudes, the smallest one $M_t = M_c$ (Figure 9a) and $M_t = 3.95$ (Figure 9b). Results show that, for the entire forecast period of 3–48 h, both the Hi-net and JMA forecasts significantly outperform the generic model and that the same result is valid in all individual forecast periods for the case of the lowest target magnitude $M_t = M_c$. Conversely, for $M_t = 3.95$, because of the small number of $m > M_t$ aftershocks, the scores tend to have large error bars and, even if the Omi method generally outperforms the generic model, this is not statistically significant for most cases (Figure 9b).

Another interesting item in Figure 9 is the comparison of the performance of the Omi method implementing the JMA catalog against the one implementing the Hi-net automatic catalogs. In general, the results show that the JMA forecast significantly outperforms the Hi-net forecast in the case of the small target magnitude $M_t = M_c$, probably because of the better accuracy of the JMA catalog. On the other hand, the two performances are comparable for $M_t = 3.95$ indicating that, even if the automatic Hi-net catalog is less accurate than the JMA catalog, it provides reasonable results for target magnitudes $M_t \geq 3.95$. This is an important result since it is the only catalog available in real time.

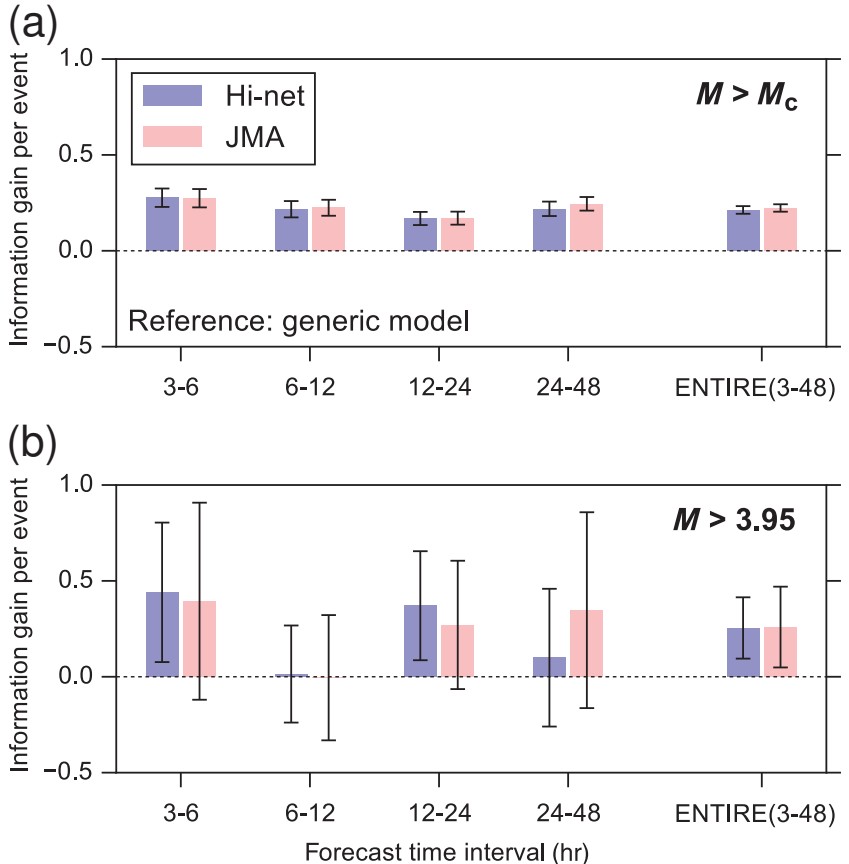

**Figure 9.** Information gain per aftershock of the forecasts based on the Hi-net and JMA catalogs, respectively, relative to the generic model for the cases with: (**a**) $M_t = M_c$; and (**b**) $M_t = 3.95$. If the lower bound of the error bar is greater than 0, the forecast is significantly better than the generic model with a probability larger than the 95%. From Omi et al. [10].

*6.2. The Lippiello Method*

Lippiello et al. [16] proposed a method based on the results presented in Section 4, which show that the instrumental envelope $\mu_e(t)$ can be reproduced by the theoretical envelope $\mu_{th}(t)$ given in Equation (15). In particular, $\mu_{th}(t)$ can be tuned to recover Equation (13) with the same parameters $\phi$ and $\Delta\mu$ of the instrumental $\mu_e(t)$. The central observation is that the value of the coefficients $\phi$ and $\Delta\mu$ which describe the logarithmic decay of $\mu_{th}(t)$ (Equation (13)) depend on the parameters $K$ and $c$ of the OU law (Equation (6)), implemented in the numerical simulation. This idea has been applied in a procedure which associates the best-fitting parameters $(\phi, \Delta\mu)$ in Equation (13), obtained from the experimental signal, to the pair $(K, c)$ used in numerical simulations of the OU law. The procedure is schematically illustrated in Figure 10. Firstly one evaluates the value of $\tau$ which is the best approximation for $\mu_e(t)$ in Equation (14) during the first 60 s. Fixing $p = 1.1$, the estimated value of $\tau$ is used to generate many numerical signals $\mu_{th}(t)$ for different choices of $K$ and $c$ according to Equation (15). Then, one compares, in the learning period $t - t_0 \in [T_1, T_2]$, the average value of the numerical signal $\mu_{th}(t)$ with the experimental one $\mu_e(t)$. The slope $\phi$ of $\mu_{th}(t)$ depends fundamentally on the $c$-value, whereas $K$ controls its vertical shift $\Delta\mu$. As a consequence, after choosing a given $K$-value, one varies the $c$-value until the slopes of $\mu_{th}(t)$ become similar to the experimental $\mu_e(t)$ (Figure 10a). The $c$-value producing this effect is then defined as $\bar{c}$ and one generates different numerical catalogs with $c = \bar{c}$ and different values of $K$ (Figure 10b). The value of $K$ minimizing the difference between $\mu_e(t)$ and $\mu_{th}(t)$ in the interval $[T_1, T_2]$ is defined as $\overline{K}$. The pair of values $(\overline{K}, \bar{c})$ is considered the best representation of experimental data and is used to forecast aftershock occurrence at times $t - t_0 > T_2$, according to Equation (6).

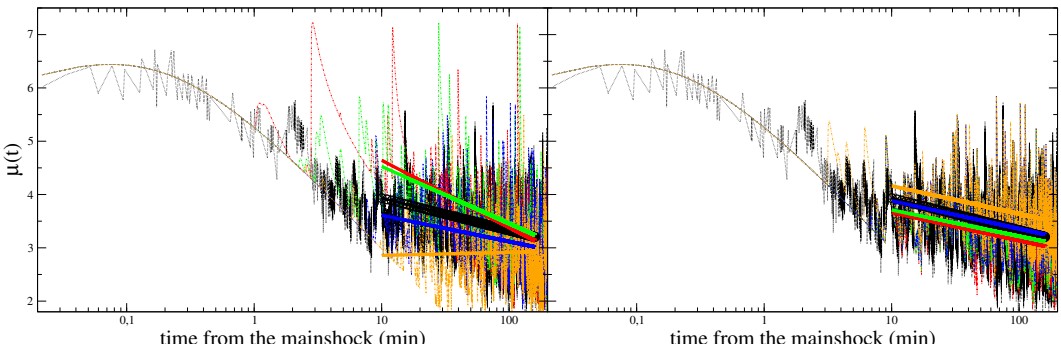

**Figure 10.** (**Left**) Black dotted-lines represent the envelope function $\mu_e(t)$ of the Lixouri earthquake in Greece recorded at the station LKD2 located at 70 km from the mainshock epicenter. Colored dot-dashed lines are used for $\mu_{th}(t)$ with $\tau = 11$ s, $K = 1.15$ and different values of $c$ ranging in the interval $[0.01, 4.5]$ days. Black circles represent the logarithmic fit (Equation (15)) in the interval $[10, 160]$ min of the experimental envelope function, whereas continuous lines are used for the best fit of the numerical $\mu_{th}(t)$, with $c$ increasing from 0.01 to 4.5 days from top to bottom. (**Right**) The same as in the left panel but plotting numerical data $\mu_{th}(t)$ with $\tau = 11$ s, $c = 1.15$ days and different values of $K \in [0.75, 2.95]$. Continuous lines are the the logarithmic fits of numerical data, in $[10, 160]$ min, with $K$ increasing from bottom to top. From Reference [16].

Test of the Procedure

To test their method, Lippiello et al. [16] considered as target aftershocks all the events producing in the envelope function $\mu_e(t)$ a peak with amplitude larger than $\mu_0 = \mu_M - 3$ and define as $N_3(t)$ their cumulative number in the temporal interval $[T_2, t - t_0]$, after the mainshock. Similarly, the number $N_2(t)$ is the cumulative number of events producing peaks larger than $\mu_0 = \mu_M - 2$.

The quantity $N_2(t)$ and $N_3(t)$ are plotted in Figure 11 for three mainshocks from three different geographic regions, for times $t > 160$ min. Lippiello et al. [16] compared the instrumental number of $N_3(t)$ and $N_2(t)$ with those expected according to the OU law (Equation (6)) after implementing the best values of $K$ and $c$ ($\overline{K}$ and $\overline{c}$) obtained according to the Lippiello procedure. More precisely, Lippiello et al. [16] considered a learning period $[T_1, T_2]$ min with $T_1 = 10$ min and different values of $T_2$. They found that for values of $T_2 \gtrsim 160$ min the estimate of $\overline{K}$ and $\overline{c}$ became quite stable. Therefore they consider $T_2 = 160$ min and found that at all times $t > T_2$ the Lippiello method predicts with reasonable accuracy the number of occurred aftershocks. Differences between predicted and observed aftershock number are typically smaller than 20% and always within the error bars. For comparison, in the same Figure 11, Lippiello et al. [16] also plotted the expected number $N_3(t)$ and $N_2(t)$ according to a generic model which implements in the OU law Equation (7) the value of $K_0, c$ and $\alpha$ obtained as average over all sequences with $m_M > 5$, recorded in Southern California [55]. We observe that the number of the predicted strong aftershocks according to this generic model is much smaller (approximately ten times) than the observed one. We wish to stress that the estimate of $K$ and $c$, for each specific sequence, on the basis of the earthquakes recorded in the official catalogs up to the time $T_2$ leads to unreliable results. As an example, in the case of the Lixouri earthquake only three earthquakes are reported in the Greek catalog in the first thirty minutes after the mainshock. The situation is a little better after the L'Aquila and Hector mine earthquake when 20 events are reported in regional catalogs in the first thirty minutes. These numbers are too small to produce a reasonable estimate of $K$ and $c$, which, in all cases, would be very biased because of the incompleteness of datasets as confirmed by the absence of earthquakes with magnitude smaller than $m = 3$, in official catalogs in the first thirty minutes.

Summarizing, results of Figure 11 clearly show that the Lippiello method performs much better than the generic model providing a reasonable aftershock forecasting. Very recently, Lippiello et al. [17] proposed a more efficient procedure, still based on the agreement between $\mu_{th}(t)$ and $\mu_e(t)$, which produces even more accurate STA forecasting.

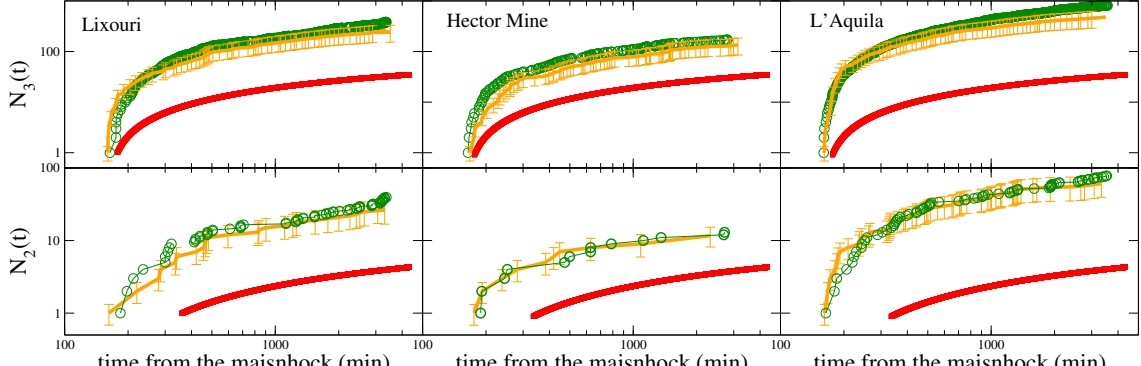

**Figure 11.** The quantities $N_3(t)$ (**top**) and $N_2(t)$ (**bottom**) are plotted for $t - t_0 > T_2 = 160$ min as green circles for the three main-aftershock sequences of Figure 1: the 26 January 2014 $m = 6.1$ Lixouri earthquake (**left**), the 16 October 1999 $m = 7.1$ Hector Mine earthquake (**middle**) and the 6 April 2009 $m = 5.9$ L'Aquila earthquake (**right**). Red squares are the expected values according to Equation (6) using the average values obtained in Reference [55]. The orange curves are the expected values using in Equation (6) the best parameters $K$ and $c$ inverted from the experimental fit of $\tau$, $\phi$ and $\Delta\mu$. The error bars in each plot incorporate both the uncertainty in the estimate of $(\Delta\mu, \phi)$ and fluctuations in the aftershock number for given values of $\overline{K}$ and $\overline{c}$. From Reference [16].

## 6.3. Comparison between the Omi and the Lippiello Methods

The Omi method needs as an earlier stage an automatic routine for real-time automatic detection. The only assumption is that the GR law holds up to the lower considered magnitude $m_c$, that is a widely accepted idea within the seismological community. Conversely, the key assumption of the Lippiello method is that the theoretical envelope (Equation (15)) reproduces the instrumental one $\mu_e(t)$. This hypothesis is less consolidated but allows one to evaluate seismic hazard directly from the envelope function $\mu_e(t)$ without any information on occurrence times, magnitude and locations of earthquakes producing the observed signal. Overcoming all problems related to event identification and location, the Lippiello method presents some advantages:

(i) It is faster. Indeed, aftershock localization is a non-trivial routine involving the elaboration of at least the seismic signal from three different seismic stations.

(ii) It works when only few events are identified by the automatic detection routine whereas the Omi et al method needs that at least ∼30 aftershocks must be identified [15].

(iii) It provides the in-situ occurrence probability by simply installing a seismic station in the site of interest. This could be particularly useful in areas with a very low dense seismic network and where automatic detection routines are not efficient.

(iv) It provides directly in output the probability of peaks of the local ground velocity and therefore it overcomes the large amount of uncertainty [56], which is present in the attenuation relations necessary to convert aftershock occurrence probability to the local ground motion intensity.

Summarizing, the two methods appear as two complementary approaches to the same problem and can be simultaneously adopted.

We finally remark that in the OMI method the spatial dependence of the aftershock occurrence probability can be easily included by multiplying the OU law (Equation (6)) for a decreasing function of the distance from the mainshock epicenter. However, the parameters controlling this decay are very difficult to be inverted from the on-going specific sequence and average quantities must be considered. On the other hand, in the Lippiello method the spatially dependence is, at least in part, implicitly considered. Indeed, the method does not give in output the probability to have a given magnitude aftershock but the occurrence probability of events which produce peaks in the envelope $\mu_e(t)$, in the position where the station is located, larger than a reference value $\mu_0$. This probability, therefore, clearly depends on the distance of the station from the mainshock epicenter.

## 7. Conclusions

In this review article, we show that, in the first part of aftershock sequence, incompleteness is an intrinsic property of seismic data. Indeed, the overlap of seismic signals makes the envelope function always greater than $\mu_{min}(t)$. This lower threshold $\mu_{min}(t)$ can be related to the minimum aftershock magnitude $m_{min}(t)$ identifiable at time $t$ since the main shock and indicates that it is feasible to obtain more accurate catalogs but it is impossible to reach completeness levels below $m_{min}(t)$. This result also provides an explanation for the dependence of $m_c(t)$ on the time elapsed from the main shock occurrence. We illustrate how the incompleteness affects the estimate of the parameters of STA forecasting models and we present some models which take it explicitly into account. In particular, we present an interpretation of the mechanisms responsible for the existence of $\mu_{min}(t)$ in terms of the overlap of coda-waves generated by each individual aftershock: The combination of the decay of the aftershock rate (OU law) with the power law relaxation of coda waves produces an envelope function $\mu_e(t)$, which, on average, depends logarithmically on the time since the main shock. We illustrate the bias induced in the estimate of model parameters because of the incompleteness of the instrumental catalog. A deeper investigation is necessary to establish a quantitative relationship between the expected error in the estimate of model parameters and the degree of incompleteness of the catalog.

We also show that the parameters of the logarithmic dependence of $\mu_e(t)$ appear strictly related to the parameters of the OU. We then describe a procedure based on this observation and developed in [16] to extract the OU law parameters from a fitting procedure applied to the experimental $\mu_e(t)$. This approach overcomes all problems related to event identification and location since seismic hazard is evaluated directly from the envelope function $\mu_e(t)$ without any information on occurrence times, magnitudes and locations of earthquakes producing the observed signal.

We also illustrate the Omi method [7,9,10,15] proposed to overcome the problems of STA forecasting caused by the incompleteness of instrumental data. We show that the method, based on the detection rate function, provides reliable aftershock forecasting on the basis of incomplete instrumental catalogs.

Summarizing, we review very recent proposals to develop real-time systems for automatic aftershock forecasting. The above procedures have been up to now tested retrospectively but appear already suitable to be implemented in prospective tests. These methods apply the OU law or the ETAS model without taking into account the spatial variability of seismicity. Future developments should correspond to space-time models providing a space dependent forecasting, particularly useful in aftershock sequences with a complex spatial distribution.

**Funding:** This research received no external funding.

**Acknowledgments:** E.P. and V.K. acknowledge support of this work by the project HELPOS, Hellenic System for Lithosphere Monitoring (MIS 5002697), which is implemented under the Action Reinforcement of the Research and Innovation Infrastructure, funded by the Operational Programme "Competitiveness, Entrepreneurship and Innovation" (NSRF 2014–2020) and co-financed by Greece and the European Union (European Regional Development Fund). Geophysics Department Contribution 000/2019.

**Conflicts of Interest:** The authors declare no conflict of interest.

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
