# Peer review of "Post Seismic Catalog Incompleteness and Aftershock Forecasting"

_geosciences, doi:10.3390/geosciences9080355_

Round 1

Reviewer 1 Report

The authors answered to my single comment in the review of the initial submission.

I recommend the manuscript for publication in Geosciences.

Author Response

We thank the referee for the further reading of our manuscript and for recommending it for publication.

Reviewer 2 Report

This manuscript well covers the topics of short-term aftershock incompleteness (STAI) and recently developed methods for aftershock forecasting taking account for STAI. These topics are important in the field. 

I would like to suggest to discuss the following issues somewhere in the manuscript.

- Short-term aftershock incompleteness has been soemtimes treated in an ad hoc way for aftershock forecasting. For example, see the supplement in the following reference.

Marzocchi, W., & Lombardi, A. M. (2009). Real‐time forecasting following a damaging earthquake. Geophysical Research Letters, 36(21).

- A significant number of early missing aftershocks can be recovered by employing a sophisticated detection algorithm such as matched filter (please see the following study). On the other hand, it is impossible to recover all the missing aftershocks even if we rely on such a sophisticated method. These studies also showed that the c-value of the modified Omori law is generally very small if the missing early aftershocks are appropriately recovered.

Peng, Z., & Zhao, P. (2009). Migration of early aftershocks following the 2004 Parkfield earthquake. Nature Geoscience, 2(12), 877.

Enescu, B., Mori, J., Miyazawa, M., & Kano, Y. (2009). Omori-Utsu law c-values associated with recent moderate earthquakes in Japan. Bulletin of the Seismological Society of America, 99(2A), 884-891.

- The aftershock forecasting methods described in this manuscript only consider the temporal forecasting. It would be worth if you discuss how to extend these methods to generate spatio-temporal forecasts.

Author Response

We thank the referee for stating that our "manuscript well covers the topics of short-term aftershock incompleteness (STAI) and recently developed methods for aftershock forecasting taking account for STAI".
We also thank the referee for the useful suggestions implemented in the revised version of the manuscript.
More precisely:

- The referee writes:
" Short-term aftershock incompleteness has been soemtimes treated in an ad hoc way for aftershock forecasting. For example, see the supplement in the following reference.
Marzocchi, W., & Lombardi, A. M. (2009). Real‐time forecasting following a damaging earthquake. Geophysical Research Letters, 36(21)."

We have added the citation to the interesting paper by Marzocchi & Lombardi at line 99 of the revised version.

- The referee writes:
"A significant number of early missing aftershocks can be recovered by employing a sophisticated detection algorithm such as matched filter (please see the following study). On the other hand, it is impossible to recover all the missing aftershocks even if we rely on such a sophisticated method. These studies also showed that the c-value of the modified Omori law is generally very small if the missing early aftershocks are appropriately recovered.
Peng, Z., & Zhao, P. (2009). Migration of early aftershocks following the 2004 Parkfield earthquake. Nature Geoscience, 2(12), 877.
Enescu, B., Mori, J., Miyazawa, M., & Kano, Y. (2009). Omori-Utsu law c-values associated with recent moderate earthquakes in Japan. Bulletin of the Seismological Society of America, 99(2A), 884-891."

We have added this information and the two references at lines 199-201 of the revised version of the manuscript.

- The referee writes:
"The aftershock forecasting methods described in this manuscript only consider the temporal forecasting. It would be worth if you discuss how to extend these methods to generate spatio-temporal forecasts."

We added a paragraph on this issue in the revised version (lines 429-437).

This manuscript is a resubmission of an earlier submission. The following is a list of the peer review reports and author responses from that submission.

Round 1

Reviewer 1 Report

The manuscript "Post seismic catalog incompleteness and aftershock forecasting" by  Eugenio Lippiello, Alessandra Cirillo, Cataldo Godano, Elefetheria Papadimitriou and Vassilis Karakostas is devoted to a serious problem of the time-dependent completeness of earthquake catalogues caused by a "shading" effect. A set of different approaches to take into account this effect in the aftershock occurrence models is reviewed. The authors obtained an important conclusion that a rise of seismic station density will not decrease the number of events missed in the catalogue after larger earthquakes.

I have just one comment. To demonstrate advantages of the method of [Lippiello et al., 2016], the authors compare predictions of the aftershock activity during about 3 days after three prominent large earthquakes, based on the learning period 10-160 min, with prediction using Reasenberg-Jones approach with average regional parameters (fig. 10). I think this comparison is not appropriate because the authors actually compare two different methods using two different data sets. More appropriate would be to use the Reasenberg-Jones with at least parameter K estimated using data from the learning period, taking regional averages for other parameters. It is important to show that taking into account a "branching shading" outperforms in forecasts taking into account shading by just the main shock.

Reviewer 2 Report

My comments are in the word-file separately.

Reviewer 3 Report

Summary The goal of this study, as outlined in the abstract, is to provide a review on methods for real time aftershock forecasting considering incomplete recordings immediately after a large earthquake. The study starts by mentioning spatial and temporal variation of the magnitude of completeness, above which recordings are assumed to be complete. Then the authors state that incomplete recordings bias certain Omori-Utsu and ETAS parameter estimates and that low network density does not lead to incomplete aftershock recordings. Afterwards a so called envelope function is addressed, which is introduced as the envelope of the seismic signal of a mainshock. In Section 5 the ETASI model is introduced which extends the ETAS model’s earthquake rate lambda with a term that accounts for the reduced aftershock rate after a large earthquake through a detection function. In the last Section the evaluation of two different ETASI models are shown. Weaknesses Unfortunately the presented study fails to review the topic outlined in the abstract which are forecasting methods for real time aftershock prediction.  As a reader I expect from a review study to summarize all relevant literature on the topic, to categorize this literature when possible into groups with common characteristics, to state pros and cons of each group and/or the methods, to conclude in which situation which method is appropriate. With this information the reader gets an overview of different methods and is able to choose which method to apply in a given situation. 1) The review is  incomplete The reader remains in the dark whether the review is exhaustive.  The authors miss to state whether the publications they consider to review are exhaustive, and if not why they choose to review only certain publications. I was confused why the authors restricted the study to only certain methods. At no point has there been a motivation for the choice of certain methods. In Section 2 the authors cite two methods to quantify spatial Mc variation, Mignan et al., 2011 and Amorese et al., 2007. It remains unclear if these two methods are the only ones to estimate spatial Mc and if there is a difference between them. The reviewer knows of at least two more publications on this topic, which is Schorlemmer and Woessner, 2008 and the CORSSA review paper on Mc (Mignan and Woessner, 2012). In Section 3 the authors mention the influence of aftershock incompleteness on Omori-Utsu and ETAS parameter estimates. They cite two studies on this topic, Utsu et al., 1995 and Zhuang et al., 2017. There exist a lot more studies on biased parameter estimates due to aftershock incompleteness, to name a few:  Helmstetter et al., 2006; Werner et al., 2011; Hainzl et al., 2013; Omi et al., 2014; Kagan, 2004; Hainzl, 2016. The authors should consider all studies or motivate why they choose to neglect them. In Section 6 the performance of forecasting methods by Omi et al., 2013; 2015; 2016 and Lippiello et al., 2016 is reviewed. What about other methods like Zhuang et al., 2017 or Hainzl, 2016? The authors should state why only these two methods are considered. 2) No new information is provided A second drawback of this review is that methods are merely quoted and at no point discussed. The review does not convey information about differences between methods, does not state pros and cons of the methods and does not conclude in which situation to apply which method. After reading this review I have not gained any new insights. In Section 2, the authors cite methods to estimate the spatially and temporally variable completeness magnitude Mc. It remains unclear under which circumstances one should be preferred over the other. In Section 6 the performance of forecasting methods by Omi et al., 2013; 2015; 2016 and Lippiello et al., 2016 is reviewed. Information about pros and cons of those methods, which method should be preferred over the other in which situation, which method performs better is not provided. 3) They sell the results of other studies as their own The most disturbing feature of this review is that whole paragraphs of other studies are recited or even copied word by word without reference. The content of those paragraphs wrongly appears to be original work of this study. In Section 5 line 224 the authors claim: “[…] we found […]” and then copy the findings of de Arcangelis et al., 2018 word by word (!) (compare their lines 222-228 with de Arcangelis et al., 2018). In Section 2 line 80-88 they copied the conclusion on some model comparison (almost word by word!) from Omi et al., 2015, without referring to it.  

Reviewer 4 Report

As a reviewer I came away with some very mixed impressions of this paper.  On the one hand, I think that this paper has some important and interesting science, and for that reason I like the paper.  One the other hand, I felt that the emphasis in the paper somewhat missed its mark.  It overstates the importance of the results that it shows, and it does not present in a clear fashion the important theoretical developments that it discusses.

The paper discusses aftershock forecasting and the problem of earthquake catalog incompleteness affecting aftershock forecasts.  In its abstract the paper refers to the “huge incompleteness” of the catalogs immediately following a mainshock, and the paper discusses “novel methods compared with traditional ones” for finding the parameters to use in aftershock forecasting models.  I believe that both of these are dramatic overstatements of the real situation.  Furthermore, I believe that this paper misses the chance to make a real contribution to the aftershock forecasting problem, as I discuss below.

First of all, let me say that I agree with the authors that the incompleteness of earthquake catalogs immediately following a strong mainshock is due to the high-amplitude coda waves of the mainshock and of strong aftershocks that occur within minutes to a few tens of minutes after a mainshock.  In many aftershock sequences that I have looked at, the strongest aftershocks often occur within the first 10-15 minutes after strong mainshocks, and these aftershocks themselves have significant coda waves that affect the seismograms for many minutes afterwards.  For the kinds of mainshocks for which the aftershock sequences are analyzed in this study (onshore events in the low Mw 7 range), aftershock catalogs typically only contain the largest events that take place during the first 30-60 minutes following the mainshock.  During this time period typical automatic event detection systems are only capable of finding larger earthquakes within coda-wave dominated ground motions that the seismic stations detect.  The automatic systems usually have not been designed to look for smaller earthquakes within such a high “noise” environment (where the “noise” is the coda wave energy).  Persons who do manual event identification and arrival time picks for event locations usually have so many aftershocks to deal with after a strong event that they concentrate more of their time on the aftershocks that take place after the strong coda waves die away.  It is for these two practical reasons that there are few aftershocks, and generally only large magnitude aftershocks, that are in catalogs during the first hour or two following an Mw 7 or larger event.  This is clearly seen in the data points in Figure 3 and Figure 6, where the density of dots (i.e., individual aftershocks) becomes large and covers a wide range of magnitudes only at times after about 0.05-0.10 days (i.e., 1.2-2.4 hours) after the mainshocks.  I agree with the authors that the coda waves are a cause of this, but there are also practical considerations of event detection and location that affect the early aftershock catalogs.

With that said in the previous paragraph, I think that after 1-2 hours after a mainshock the number of aftershocks and their magnitude distribution becomes complete enough that estimates of the various aftershocks parameters necessary for aftershock forecasting (no matter which forecasting model is used) can start to be made.  It is not clear to me that a catalog must be 100% complete before reliable estimates of the parameters needed for an aftershock forecasting model can be found.  For example, can parameters that are reliable enough for an aftershock forecast be determined using a catalog that is only 90% complete?  Or using a catalog that is only 80% complete? Or only 50% complete?  The analysis in Section 3.1 and in Figure 4 from Zhuang et al. shows how magnitude completeness affects the estimation of ETAS parameters.  I think this is an important modeling result.  I would guess that the aftershock catalog for the 2016 Kumamoto earthquake is likely complete down to M 3.0 starting very early in the aftershock sequence (perhaps within 1 hour after the mainshock origin time).  The results in Figure 4 suggest that good estimates of the parameters c and p can be made using events of M1.5 and larger, even though this is likely well below the threshold of completeness.  A good estimate of mu might be made with events above M2.0.  Only the parameters K, A and alpha appear to need a complete catalog (i.e., events of M≥3) for their proper estimation.

I like the analyses in Sections 4 and 5, where the envelope function is described and the ETASI model is developed.  I think this is a good way to model the early parts of an aftershock sequence (i.e., the first 1-2 hours) when the seismograms are dominated by strong coda waves and therefore event detection is difficult.  Section 6 tests the Omi method and the Lippiello method for aftershock forecasting during the first minutes to hours after a mainshock.  I am not surprised that these methods are better than the generic model because the generic model is simply some kind of average model for all aftershock sequences.  Any adaptive method like the two described in Section 6 should perform better that a generic model.  I think the missed opportunity in this paper is the chance to show how the accuracy of aftershock forecasts can be improved with time after a mainshock and to show the time when the aftershock forecasts become reasonably reliable (for example, 90% or 95% accurate).  This is purportedly shown in some manner in Sections 6.2 and 6.2.1, but I simply could not follow those sections well enough to figure out what they were saying.  If I interpret Figure 9 correctly, it seems to say to me that after about 2 1/2 hours (160 minutes) the estimates of the parameters in the Lippiello et al. model are good enough that accurate aftershock forecasts can be made using those parameter estimates (although there never is a good definite of an accurate aftershock forecast anywhere in the paper).  I was not able to see the accuracy of the Lippiello et al. method myself by studying Figure 9 (too hard to figure out from the text and figure), but the text on page 15 (lines 348-350) seems to tell me the method is accurate after about 160 minutes.  This section could represent a real and important contribution in this paper, but in its current version the section is too difficult for a knowledgeable reader like myself to figure out.  It would be nice if this paper discussed the practical methods and implications of short-term aftershock forecasting during the first few hours (2-3 hours) after a mainshock.  Although not something that I would require in a paper like this, I would be curious to know the authors’ opinion about the implementation of an aftershock forecasting method as part of an automated, real-time event detection and location system.

As can be seen in my comments in a marked-up version of the paper, I have many other criticisms of the writing in this version of the paper.  In particular, there are too many variables that are undefined, as well as some variables that may be the same or may be different (C0 versus c0 versus c, for example--I could not tell the difference between these).  I have suggested improvements to the written English in a number of places to assist the authors.  For example, in seismological English we use “aftershock” as one word as well as the word “mainshock”.  The authors have used the word “aftershock” throughout the text but have split the word “main shock”.
